# Ice Mass Balance in Liaodong Bay: Modeling and Observations

**Yuxian Ma** [1,2,3], **Dewen Ding** [1,2], **Ning Xu** [2], **Shuai Yuan** [2] **and Wenqi Shi** [2,*]

1    College of Environmental Science and Engineering, Ocean University of China, Qingdao 266100, China
2    National Marine Environmental Monitoring Center, Dalian 116023, China
3    State Key Laboratory of Coastal and Offshore Engineering, Dalian University of Technology, Dalian 116024, China
*    Correspondence: wqshi@nmemc.org.cn

**Abstract:** During the winters of 2009/2010 and 2020/2021, observations were carried out at an eastern port of Liaodong Bay to examine the variations in sea ice thickness and atmospheric conditions. The daily ice thickness (DIT) and the cumulative ice thickness (CIT) are the two main observation items related to the thickness of sea ice. For DIT, the sea ice thickness gradually decreases as the temperature increases, and the freezing rate $a$ is 1.48 cm/(°C·d)$^{1/2}$. For CIT, when the temperature is $-12\,°C$, the maximum growth rate of ice thickness decreases from 3.5 cm/d to 1.5 cm/d as the ice thickness increases from 0 to 20 cm. The residual method was applied to calculate the oceanic heat flux, which is an important parameter of ice modeling, and both the analytic model (Stefan's law) and numerical model (high-resolution thermodynamic snow-and-ice model) were utilized in this work. It was found that the accuracy of the simulation results was high when the growth coefficient of the analytic mode was 2.3 cm/(°C·d)$^{1/2}$. With an oceanic heat flux of 2 W·m$^{-2}$, the maximum error of the numerical model approached 60% in 2010 and 3.7% in 2021. However, using the oceanic heat flux calculated in this work, the maximum error can be significantly reduced to 4.2% in the winter of 2009/2010 and 1.5% in 2020/2021. Additionally, the oceanic heat flux in Liaodong Bay showed a decreasing trend with the increase in ice thickness and air temperature.

**Keywords:** Liaodong Bay; sea ice thickness; Stefan's law; HIGHTSI; oceanic heat flux





## 1. Introduction

Global climate change is altering the movement of salts, gases, and nutrients in the ocean–ice–atmosphere system [1–5]. Over the past 70 years, the severity of sea ice in the Bohai Sea has weakened [6–10], and the increased volatility has led to a greater risk of sea ice hazards [11,12]. In order to clarify the relationship between climate change and sea ice in the Bohai Sea and improve the prediction of sea ice thickness, it is important to carry out both direct observations and numerical simulations of sea ice mass balance in this area [13–15].

At present, most observations and numerical simulations of the ice mass balance process focus on the polar and sub-polar regions [16,17]. Researchers observed the ice thickness, ice temperature, and ice salinity of sea ice during both the ice growth and melting periods [18], and the collected data were used to optimize the classical sea ice thermodynamic model [19]. In addition, the data were used to determine the parameters of thermodynamic modes. It was found that the oceanic heat flux plays a significant and important role in the accurate simulation of ice, and the oceanic heat flux has different characteristics depending on region and season [20,21]. However, due to climate change, the Arctic sea ice is gradually shifting from multi-year ice to first-year ice [22], and the research priority will become seasonal sea ice in the future. As the southern boundary of the frozen sea area in the Northern Hemisphere, the research on Bohai Sea ice can be an important reference for Arctic first-year ice. At present, the research works on Bohai Sea ice have mainly focused on the ice extent in recent years; for example, Zhang [23] used machine

learning methods to develop a novel empirical model with the aim to predict the sea ice area. However, the thermodynamic process of landfast ice has not been carried out. The last thermodynamic observations of landfast ice can be traced to the winter of 1989/1990, when China and Finland conducted field surveys of meteorology, sea ice temperature, and thickness in the Bayuquan Port, and obtained 8 days of continuous hourly data [24]. Based on these data, Cheng [25,26] carried out the test of the high-resolution thermodynamic snow-and-ice model (HIGHTSI), in which the oceanic heat flux was chosen as a constant 5 W/m$^2$. Because the oceanic heat flux has its own characteristics in each sea area, Ji and Yue used floating ice thickness and meteorological data collected from the Liaodong Bay JZ20-2 platform and calculated the oceanic heat flux in the 1997/1998 ice season. It was found that the largest value was 200 W/m$^2$ during the initial ice period, which then decreased to 0 during the melting period [27]. The disadvantage is that the observed data of Ji and Yue were affected by dynamic factors, and we are still lacking knowledge about the oceanic heat flux of landfast ice, although it has been identified to have an important influence on the numerical simulation of sea ice in Liaodong Bay [28–32]. Therefore, the continuous mass balance observation of landfast ice in the Bohai Sea is of great significance for seasonal sea ice research.

In this study, the relationship between sea ice thickness and air conditions was observed in Jiangjunshi Port during the winters of 2009–2010 and 2020–2021. The residual method was applied to calculate the oceanic heat flux and to analyze the factors that affect the flux. Furthermore, based on the observed sea ice thickness and meteorological data as inputs, the sea ice growth and decay were evaluated using both an analytic model and numerical simulation.

## 2. Observation Area and Data

### 2.1. Observation Area

The state of sea ice in the Bohai Sea is influenced by global climate change, and Liaodong Bay usually has the maximum ice conditions in terms of ice extent, thickness, concentration, and duration [33]. Meanwhile, ice can be observed every winter season in Liaodong Bay.

Prior to carrying out air–ice–water observations in Liaodong Bay, the primary challenge was to locate a suitable landfast ice site that meets specific criteria. The selected area must allow seawater to flow under the sea ice while remaining stationary under the ocean current. Accordingly, Jiangjunshi Port, which has severe ice conditions, was selected as the observation site. Figure 1 shows the location of Jiangjunshi Port in Liaodong Bay, with coordinates of 39°55′7.72″ N, 121°40′40.77″ E. There is a small port that is approximately 0.16 km$^2$ in Jiangjunshi Port. The water depth is 8 m, and the seawater salinity is 28‰. This location is ideal for observing the thermodynamic process of sea ice in Liaodong Bay, as it is sheltered from the dynamic factors such as currents and waves. Usually, the ice season begins in December and extends until the end of March [34].

### 2.2. Sea Ice Thickness and Atmospheric Conditions in the Winter of 2009–2010

Atmospheric data were collected during the winter of 2009–2010 using the automatic meteorological equipment positioned 10 m above the ice surface. The collected elements included wind speed, wind direction, air temperature, humidity, and pressure, with data recorded every 10 min. The measurement accuracies of wind speed, wind direction, and air temperature were 0.01 m/s, 0.01 degrees, and 0.01 °C, respectively. The observation period covered the entire ice season, as presented in Figure 2. On 5 January, the air temperature hit a lowest value of −17.99 °C.

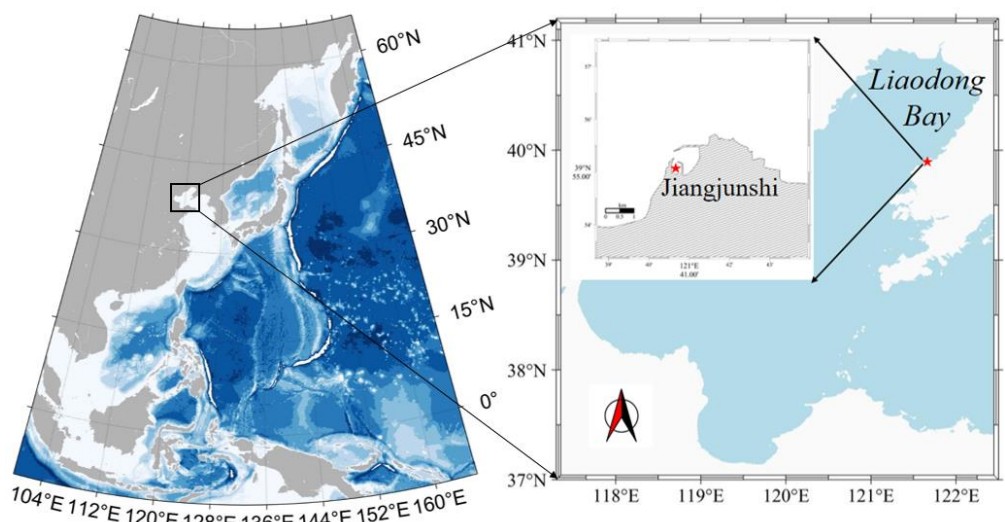

**Figure 1.** Locations of landfast ice survey site in Jiangjunshi Port.

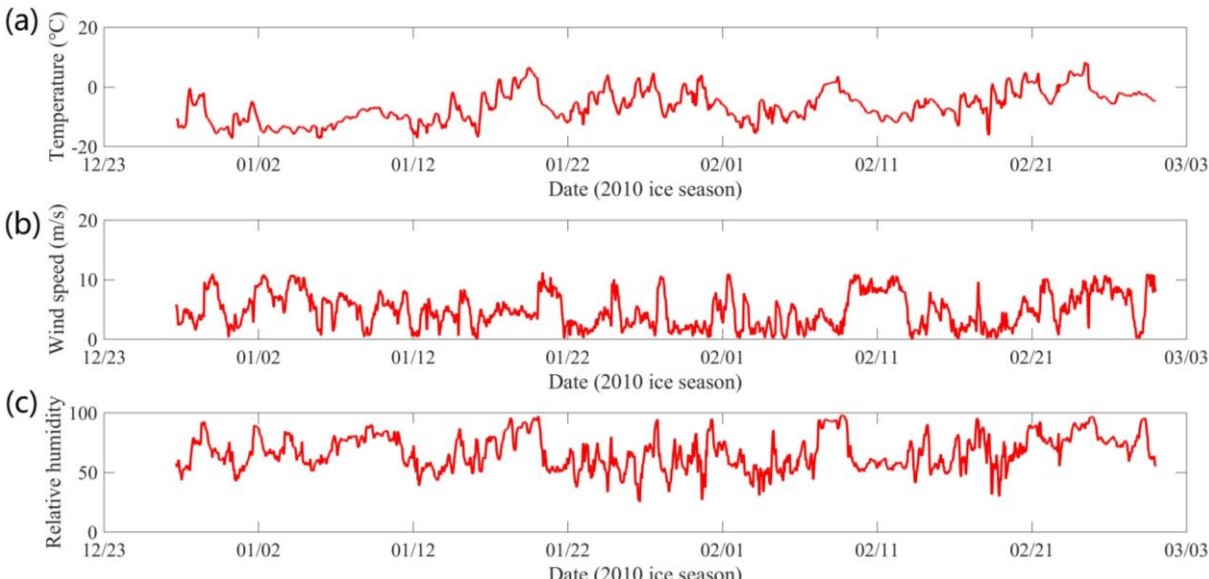

**Figure 2.** Time series (2009/2010) of (**a**) hourly measured air temperature, (**b**) hourly measured wind speed, (**c**) hourly measured humidity.

During the 2010 ice season, two sea ice observation options were available at the port covered with landfast ice. The following observation steps were taken: (1) Two areas of $2 \times 2$ m were cut into the landfast ice, and the ice within these areas was removed. To ensure that the two areas did not affect each other, they were placed 50 m apart and named ice zone 1 (IZ1) and ice zone 2 (IZ2), respectively. (2) The growth thickness was observed in IZ1 for 24 h. After measuring the ice thickness each day, the thickness was recorded as H1 and then all the sea ice in IZ1 was removed. (3) The CIT was observed in IZ2, which was measured at 9 a.m. every day and recorded as H2.

In addition, it was observed that the sea ice occasionally broke due to the tide. The broken sea ice drifted out of the port, and it interrupted the observation of sea ice thickness. According to our observations, the first period of interruption occurred from 11 January to 19 January 2010 for a total of 9 days, with an average air temperature of −7.9 °C. The second period of interruption occurred from 23 January to 15 February 2010 for a total of 24 days, with an average air temperature of −5.8 °C. Throughout the observation period, the temperature remained low, and the sea ice growth was rapid.

### 2.3. Sea Ice Thickness and Atmospheric Conditions in the Winter of 2020–2021

Atmospheric conditions were also observed during the observation period using automatic meteorological equipment. The installed sensor was the MaxiMet series GMX500, and the observed elements included air temperature, wind speed, wind direction, humidity, and air pressure. The measurement resolution for wind speed was 0.01 m/s, for wind direction it was 0.01°, for humidity it was 1%, and for temperature it was 0.01 °C. The data were recorded every 5 s, and the recording period was from 27 December 2020 to 1 March 2021. Figure 3 presents the time histories of those meteorological data. The wind speed data contain some missing parts, which were caused by the heavy frozen state of the sensor. In the third section of the model calculation, the missing data are replaced by ERA5 reanalysis data.

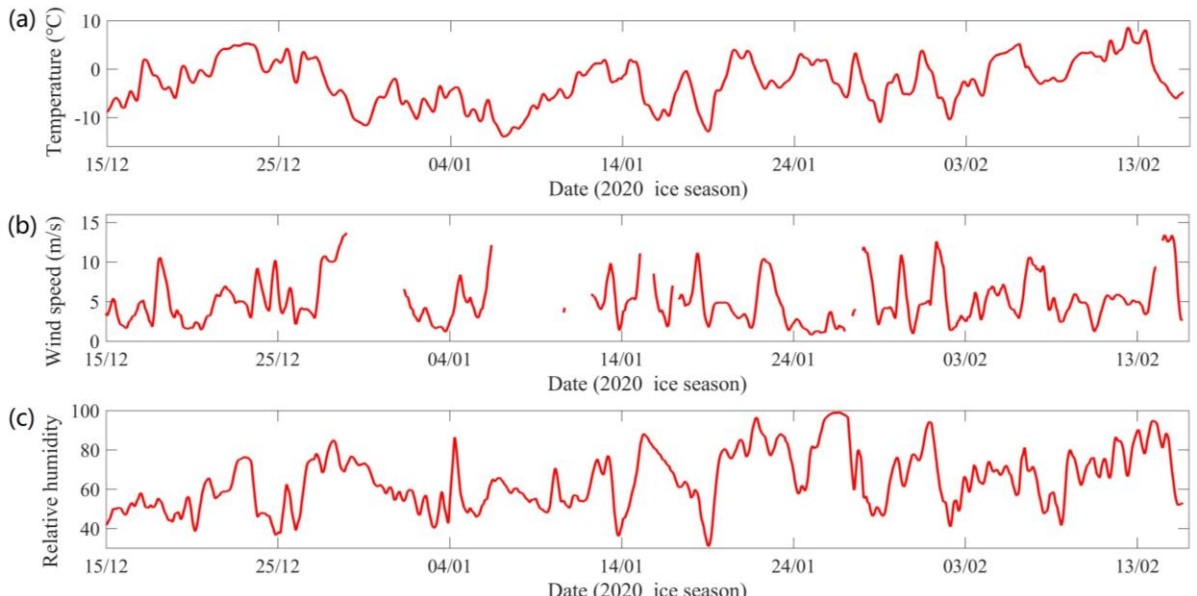

**Figure 3.** Time series (2020/2021) of (**a**) hourly measured air temperature, (**b**) hourly measured wind speed, (**c**) hourly measured humidity.

On 31 December 2020, the sea ice entered the port and froze completely. On 3 January, the measurement of ice thickness started. The measurement was conducted using an ice ruler, and the thickness was observed at 8:00 and 17:00 during the ice period. On 4 February, the tide caused the sea ice to break and the measurement of ice thickness was completed.

## 3. Methods

### 3.1. Thermodynamic Model of Sea Ice

3.1.1. Stefan's Law of Ice Growth

There are two types of thermodynamic models for sea ice: an analytical model and numerical simulation. For the analytical model, Stefan established a formula to calculate ice thickness, which only considers the heat balance at the ice–water interface. The heat released from the freezing occurring at the bottom of the ice layer is transferred to the surface of the ice layer through a constant temperature gradient. This formula is based on 4 basic assumptions: (1) under rapid temperature changes, the lag in ice temperature change is ignored; (2) the solar radiation absorbed by ice is ignored; (3) the heat flux at the bottom of the ice is ignored; and (4) the surface temperature is a function of time [35–37]. The calculation formula is:

$$\rho_i L_f \frac{dh_i}{dt} = \frac{k_i \left( T_f - T_0 \right)}{h_i} \tag{1}$$

where $k_i$ is the thermal conductivity of the ice; $\rho_i$ is ice density; $L_f$ is freezing latent heat; $T_f$ is the freezing temperature; and $T_0$ is the ice surface temperature. According to Formula (1), the ice thickness can be calculated. The initial ice thickness is $h_0$ at the initial time $t = 0$, and both sides of Formula (1) are integrated simultaneously:

$$h_i = \sqrt{h_0^2 + a^2 FDD} \tag{2}$$

$$a = \sqrt{2k_i / \rho_i L_f} \tag{3}$$

$$FDD = \int_0^t \left( T_f - T_0 \right) \tag{4}$$

3.1.2. High-Resolution Thermodynamic Snow-and-Ice Model (HIGHTSI)

Most numerical models of sea ice were developed by combining the principles of energy balance and heat conduction. In this study, the high-resolution thermodynamic snow-and-ice (HIGHTSI) model [38–43] proposed by Maykut and Untersteiner [44] was applied to simulate the growth of sea ice thickness. The HIGHTSI model follows the classical one-dimensional sea ice model, and its core is the partial differential thermal conductivity equation which considers the vertical heat and mass balance through the snow–ice–ocean system. The HIGHTSI model has been widely applied to simulate the ice thermodynamics in various locations, such as, e.g., the Bohai Sea, Antarctic sea, Arctic sea, Baltic Sea, and Finnish lakes. The key processes include the surface heat and mass balance (Equation (5)), the snow and ice temperature (Equation (6)), and the ice bottom heat and mass balance (Equation (7)):

$$(1 - \alpha_{i,s})Q_s - I(z)_0 + \varepsilon Q_d - Q_b\left(T_{sfc}\right) + Q_h\left(T_{sfc}\right) + Q_{le}\left(T_{sfc}\right) + F_c\left(T_{sfc}\right) - F_m = 0 \tag{5}$$

where $Q_s$ is the downward solar radiation for all sky conditions; $\alpha$ is the surface albedo; $I(z)$ is the solar radiation penetrating below the surface layer; $Q_d$ and $Q_b$ are the downward and upward longwave radiation under all sky conditions; $\varepsilon$ is surface emissivity; $Q_h$ and $Q_{le}$ are turbulent sensible and latent heat fluxes; $F_c$ is the conductive heat flux of the surface layer; $F_m$ is the surface melting of snow or ice; and $T_{sfc}$ is surface temperature.

$$(\rho c)_{i,s} \frac{\partial T_{i,s}(z,t)}{\partial t} = \frac{\partial}{\partial z} \left( k_{i,s} \frac{\partial T_{i,s}(z,t)}{\partial z} - q(z,t) \right) \tag{6}$$

where $T$ is temperature; $t$ is time; $z$ is the vertical coordinate below the surface; $\rho$ is density; $c$ is specific heat; $k$ is thermal conductivity (function of $T_i$ and $s_i$); $q(z,t)$ is the absorbed solar radiation below the surface layer; and the subscripts $s$ and $i$ are snow and ice, respectively.

$$\rho_i L_i \frac{dh_i}{dt} + F_w = \left( k_i \frac{\partial T_i}{\partial z} \right)_{bot} \tag{7}$$

where $h_i$ is sea ice thickness; $L_i$ is the latent heat of fusion; and $F_w$ is the oceanic heat flux.

The inputs of the mode are wind speed (*Va*); temperature (*Ta*); relative humidity (*Rh*); precipitation; and solar radiation (parameterized). The outputs of the mode are the time series of ice thickness ($h_i$) and ice temperature.

*3.2. Statistical Method*
3.2.1. Least Squares Method

The least squares method is commonly used for parameter estimation, system identification, and prediction. In this study, the least squares method was used to estimate the

parameters in the relationship between the sea ice thickness and the air temperature data. According to the least squares method, the minimization value is:

$$L = min \sum_{n=1}^{\max(n)} \left( y(n) - y(n)' \right)^2 \tag{8}$$

where $y(n)'$ is the result of sea ice thickness to be fitted, $n$ is air temperature, and $y(n)$ is the observed data of ice thickness.

### 3.2.2. Coefficient of Determination ($R^2$)

The coefficient of determination is a statistical measure of the goodness of fit. It represents the proportion of variance in the dependent variable that can be explained by the independent variables included in the model. The value of $R^2$ ranges from 0 to 1, where a higher value indicates a better fit between the observed data and the model.

$$R^2 = 1 - RSS/TSS \tag{9}$$

where $RSS$ is the sum of squared residuals and $TSS$ is the total sum of squares.

## 4. Results

Sea ice thickness is an important characteristic of sea ice severity in Liaodong Bay, and its accurate evaluation is crucial for biogeochemical cycle research. The thickness of sea ice is affected by various factors, such as atmospheric conditions, solar radiation, and oceanic heat flux. Previous studies have demonstrated that the temperature is the main factor affecting the sea ice severity in Liaodong Bay. Against this backdrop, this study analyzed the changes in sea ice thickness under meteorological effects.

### 4.1. Sea Ice Thickness Analysis Based on Stefan's Law

#### 4.1.1. Statistical Law of Sea Ice Growth Rate and Temperature

During the growth and melting of sea ice, the air temperature was found to have a significant influence on the sea ice's temperature profile. When the air temperature drops rapidly, the temperature profile of sea ice shows a linear distribution along the ice; meanwhile, when the air temperature becomes high, the ice temperature is low in the middle but high on the surface layer and bottom layer. Therefore, the growth rate of sea ice only shows statistical regularities during the period of rapid growth. Hence, the relationship between sea ice thickness and atmospheric temperature was evaluated only for the winter of 2009–2010.

Based on the observation area IZ1, the analysis focused on the daily ice thickness (DIT), which represents the thickness of sea ice growth starting from 0 cm over a 24 h period. To evaluate the growth law of DIT, Equations (2)–(4) were used to calculate sea ice thickness, where $h_0 = 0$, $h_i = a \sqrt{FDD}$, and $a$ represents the freezing rate. Based on these equations, the relationship between DIT and daily average temperature was plotted (Figure 4). The regression analysis equation is:

$$h_i = 1.48 \sqrt{FDD} \tag{10}$$

where $FDD$ is the cumulative temperature (°C·d) and $h_i$ is the thickness of sea ice (cm). The coefficient of determination is 0.38.

The results indicate that as the temperature increases, the DIT of sea ice decreases, with a freezing rate of 1.48 cm/(°C·d)$^{1/2}$. For instance, at an average temperature of $-12$ °C, the average DIT was 5.3 cm, whereas at an average temperature of $-2$ °C, the average DIT was about 2.5 cm. Additionally, there is still an increase in ice thickness of 2 cm at temperatures close to 0 °C, which may be attributed to the varying sea temperatures on different days. Notably, the freezing rate of DIT is significantly smaller than the theoretical value of 3.3 cm/(°C·d)$^{1/2}$.

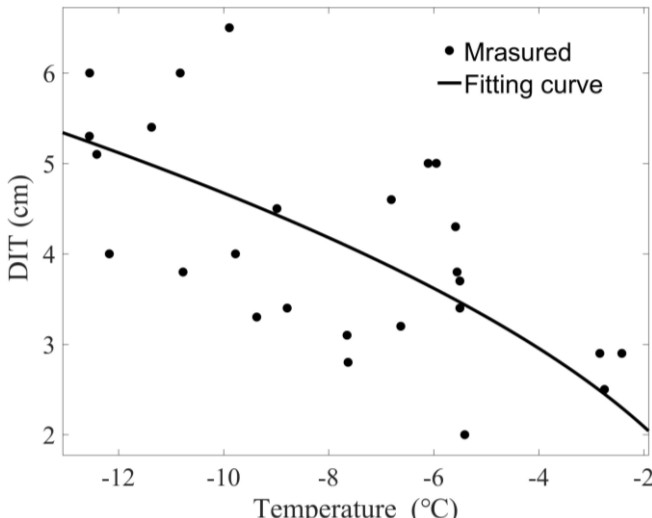

**Figure 4.** Statistical relationship between DIT and temperature.

In addition to DIT, the cumulative ice thickness (CIT) of sea ice, which was measured every day at 9 am, is another important observational parameter to consider. In Liaodong Bay, it was observed that the sea ice growth is significantly dominated by the cold fronts from Siberia. When the cold air reaches the ice area, the low temperature moves downward in the form of a cold front. It freezes the surface of sea water under the sea ice. As the cold air leaves, both the atmospheric temperature and the ice temperature increase, leading to a stable trend of increasing sea ice thickness. Obviously, the increase in sea ice thickness is affected by the amount of heat consumed by the cold front reaching the bottom of the ice. As the sea ice thickness increases, the same cold front with the same energy intensity can only cause a smaller increase in ice thickness.

To clarify the influence of air temperature and sea ice thickness on the sea ice growth rate, a three-dimensional fitting was performed, and the relationship between the average temperature, sea ice thickness, and sea ice growth rate (cm/d) was plotted, as shown in Figure 5. It can be seen that at a temperature of −12 °C, the maximum growth rate of ice thickness decreases from 3.5 cm/d to 1.5 cm/d as the ice thickness increases from 0 to 20 cm. The regression analysis equation is:

$$dh/dt(T,h) = 1.643 - 0.158T - 0.101h \qquad (11)$$

where $T$ is the temperature (°C), $h$ is the thickness of sea ice (cm), and $dh/dt$ is the growth rate of sea ice (cm/d). The coefficient of determination is 0.65.

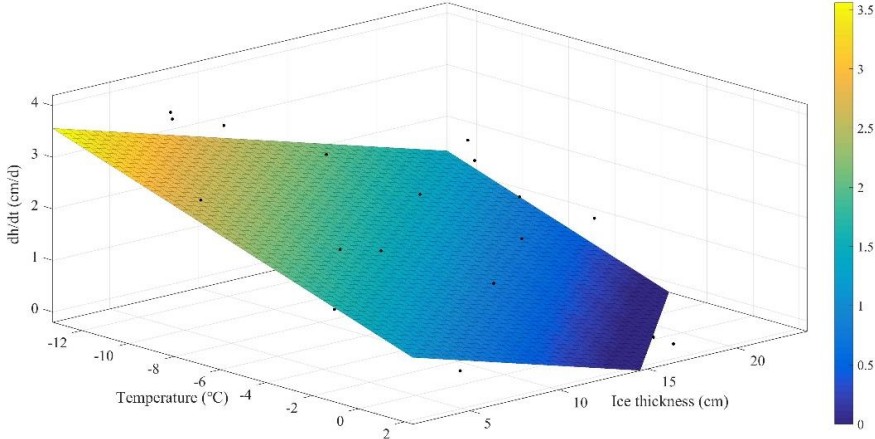

**Figure 5.** Statistical relationship between growth rate, ice thickness, and average daily temperature.

### 4.1.2. The Sea Ice Freezing Rate

In Stefan's law, the most important factor is the freezing rate. Based on Equations (2)–(4), the freezing rate is $h_i / \sqrt{FDD}$. For Jiangjunshi Port, the DIT and air temperature were used to calculate the freezing rate of first-day ice growth. The results showed that the freezing rate increased with higher atmospheric temperature and thicker sea ice (Figure 6). The regression analysis equation is:

$$a = 1.459 + 0.170\,T + 0.394\,h \tag{12}$$

where $T$ is temperature and $h$ is DIT. The coefficient of determination is 0.812.

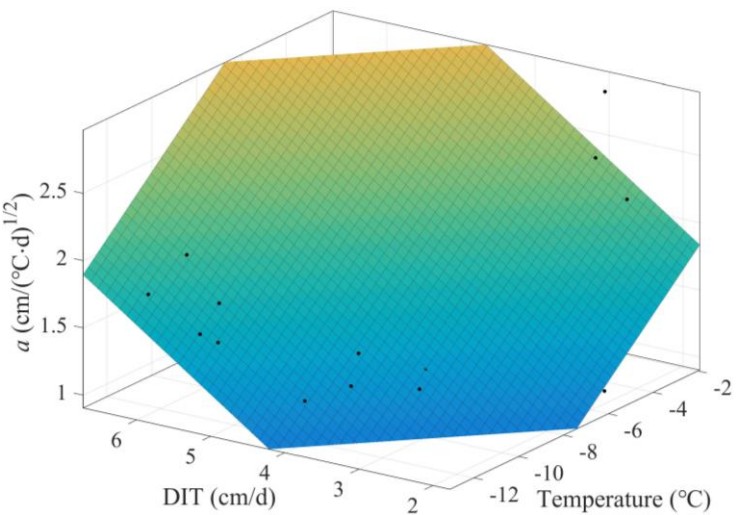

**Figure 6.** The freezing rate varies with DIT and temperature.

An accurate freezing rate is important for assessing the cumulative growth thickness of sea ice. To calculate the freezing rate of the CIT, the observation data from 23 January to 15 February 2010 were utilized. The data from 12 January to 19 January 2010 were not selected due to a short observation period. The initial ice thickness was 25 cm from 4 January to 4 February 2021. The freezing rate for each day was calculated by using $h_i / \sqrt{FDD}$, and the results are presented in Figure 7. Apparently, except for the two days when the sea ice started to grow, the freezing rate remained stable, with an average of 2.3 cm/(°C·d)$^{1/2}$.

### 4.1.3. Stefan's Law of Ice Growth

When the integration time step $dt = 1$ d, $FDD$ represents the cumulative freezing degree days. $k_i$ is 2.03 W/(m·K); $\rho_i$ is 917 kg/m$^3$; $L_f$ is 333.4 kJ/kg; $T_f$ is the freezing temperature ($-1.4$ °C); and $T_0$ is the ice surface temperature, which is approximately equal to the air temperature. According to Equation (3), the theoretical value of $a$ is 3.3 cm/(°C·d)$^{1/2}$, and the unit is calculated from the units of $k_i$, $\rho_i$, and $L_f$. The oceanic heat flux in this study was calculated based on the residual method, which was ineffective to optimize the calculation model based on the same principle. Based on the background, the theoretical values of 1.8 cm/(°C·d)$^{1/2}$, 1.71 cm/(°C·d)$^{1/2}$, 2.3 cm/(°C·d)$^{1/2}$, and 2.7 cm/(°C·d)$^{1/2}$ were used to simulate the thickness of sea ice.

Since Stefan's law assumes a linear ice temperature profile, this method is mainly suitable for the period of rapid sea ice growth caused by low temperatures. Thus, the observation data from the winter of 2009–2010 were used for the simulation, as shown in the results in Figure 8. Comparing with the measured data, the simulation results show that the ice thickness is significantly overestimated when the growth coefficient is 3.3 cm/(°C·d)$^{1/2}$ and 2.7 cm/(°C·d)$^{1/2}$, and significantly underestimated when the growth coefficient is 1.71 cm/(°C·d)$^{1/2}$ and 1.8 cm/(°C·d)$^{1/2}$. Based on the measured data and

cumulative freezing days, a sea ice growth rate of 2.3 cm/(°C·d)$^{1/2}$ is recommended for Jiangjunshi Port, and the corresponding calculation results are shown in Figure 8. The calculation errors were analyzed and presented in Table 1. The results indicate that when the growth coefficient is 1.71 cm/(°C·d)$^{1/2}$, the calculation error is over 18%; when it is 1.8 cm/(°C·d)$^{1/2}$, the error is over 13.4%; and when it is 2.7 cm/(°C·d)$^{1/2}$, the error is over 16.4%. However, when the growth coefficient is 2.3 cm/(°C·d)$^{1/2}$, except for in the initial stage, the calculation error is less than 2.8%. The theoretical value of 3.3 cm/(°C·d)$^{1/2}$ yields a calculation error of over 42.2%.

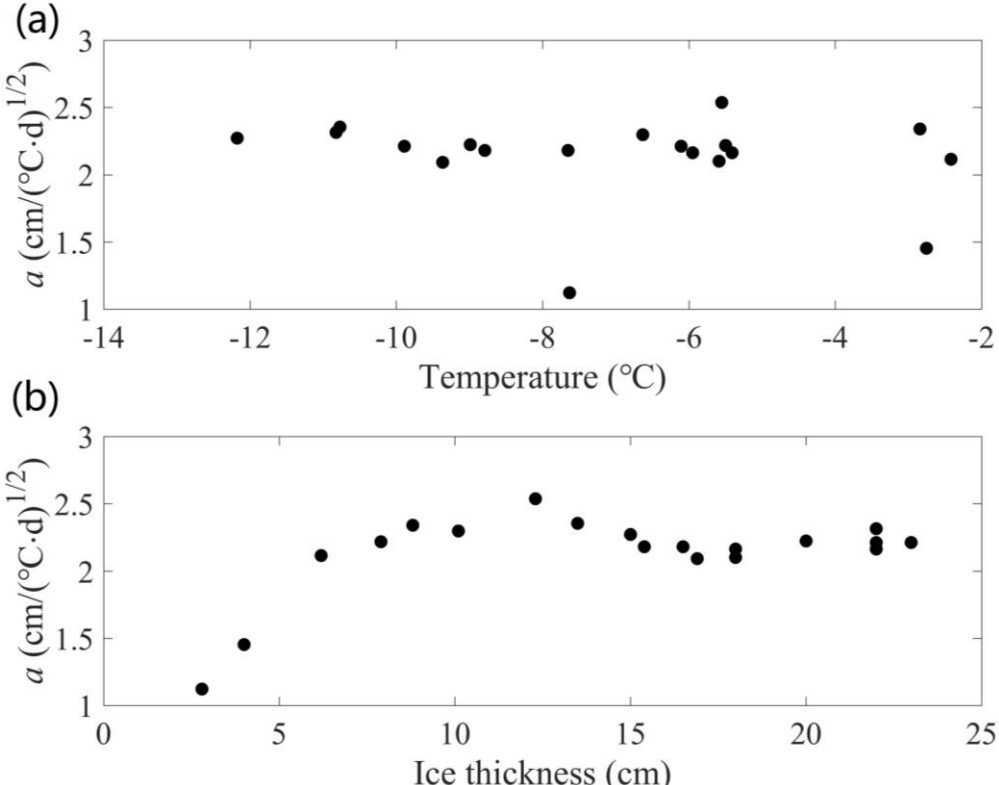

**Figure 7.** (**a**) The relationship between temperature and freezing rate; (**b**) relationship between ice thickness and freezing rate.

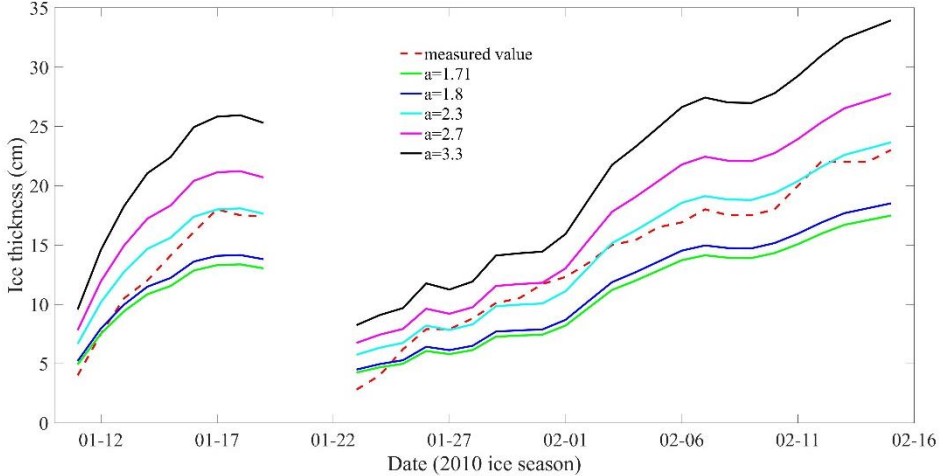

**Figure 8.** Simulation results under different growth coefficients.

**Table 1.** Simulation error at different growth coefficients.

| | a = 1.71 | | a = 1.8 | | a = 2.3 | | a = 2.7 | | a = 3.3 | |
|---|---|---|---|---|---|---|---|---|---|---|
| | Difference | Error | Difference | Error | Difference | Error | Difference | Error | Difference | Error |
| SD1 *-15 January | −2.56 | 18.1% | −1.88 | 13.4% | 1.51 | 10.7% | 4.22 | 30.0% | 8.30 | 58.8% |
| SD1-19 January | −4.37 | 25.1% | −3.61 | 20.7% | 0.22 | 1.3% | 3.29 | 18.9% | 7.89 | 45.3% |
| SD2-27 January | −2.11 | 26.7% | −1.77 | 22.4% | −0.07 | 0.8% | 1.30 | 16.4% | 3.34 | 42.3% |
| SD2 *-6 February | −3.19 | 18.9% | −2.38 | 14.1% | 1.65 | 9.7% | 4.87 | 28.8% | 9.71 | 57.5% |
| SD2-11 February | −4.93 | 24.7% | −4.05 | 20.2% | 0.38 | 1.9% | 3.93 | 19.6% | 9.24 | 46.2% |
| SD2-15 February | −5.52 | 24.0% | −4.50 | 19.5% | 0.65 | 2.8% | 4.76 | 20.7% | 10.92 | 47.5% |

Note(s): * SD1 is 12 January–19 January in winter 2009–2010; SD2 is 23 January–15 February in winter 2009–2010.

It was found that the calculation errors were due to two main factors: the negative accumulated temperature in the early stage and the different growth coefficients at different locations. The growth coefficient was measured to be 1.8 cm/(°C·d)$^{1/2}$ in the waters of Huludao on the west coast of Liaodong Bay, 1.71 cm/(°C·d)$^{1/2}$ in the northern waters of Liaodong Bay, and 2.7 cm/(°C·d)$^{1/2}$ on the platform in Liaodong Bay [32]. For the negative accumulated temperature in the early stage, Cao simulated lake ice using Stefan's law and found that predicting ice thickness two weeks in advance was more accurate [36]. In the case of Jiangjunshi Port on the east coast of Liaodong Bay, the measurements showed that the sea ice started to grow from 0 cm, and the growth coefficient given in this study is 2.3 cm/(°C·d)$^{1/2}$.

*4.2. High-Resolution Thermodynamic Snow-and-Ice Model (HIGHTSI)*

4.2.1. Oceanic Heat Flux

In 1982, McPhee and Untersteiner proposed a method for calculating oceanic heat flux using the sea ice energy balance, which relies on the observed data of the sea ice mass balance and temperature profile [45]. This method only requires data of the thermodynamic parameters of sea ice, such as ice bottom position, ice temperature, and ice salinity. In cases where the observation data of the sea ice temperature are not available, the oceanic heat flux can still be calculated by combining the thermodynamic numerical model with the measured thickness data of sea ice. This method is called the residual method and is also known as the method of measuring oceanic heat flux using ice thickness. The residual method has been extensively applied to calculate the oceanic heat fluxes under the pack ice in east Antarctica [46,47], under the pack ice in the Alaskan Beaufort Sea [48], and under the landfast ice in McMurdo Sound [49]. The relationship between the energy balance of ice bottom ablation or accretion can be expressed as:

$$F_w = F_c + F_L \tag{13}$$

where $F_w$ is oceanic heat fluxes; $F_c$ is the conductive heat flux; and $F_L$ is the latent heat flux. The sign convention is that upward and melting are positive, whereas downward and freezing are negative. The calculation formula of conductive heat flux and latent heat flux can be expressed as:

$$F_c = k_i \cdot \partial T_i / \partial z \tag{14}$$

$$F_L = -\rho_i \cdot L \cdot \partial z_i / \partial t \tag{15}$$

where $(\partial T_i / \partial z)$ is the temperature gradient in ice; $(\partial z_i / \partial t)$ is the growth rate of sea ice; and the thermal conductivity, latent heat of fusion, and density of the ice are $k_i$, $L$, and $\rho_i$, respectively, with values of 2.03 W/(m·°C), 333.4 kJ/kg, and 917 kg/m$^3$.

Oceanic heat fluxes were calculated for three different time periods (twice in 2010 and once in 2021). Sea ice surface temperature was similar to the air temperature, where the freezing point of sea ice is −1.4 °C. The temperature profile in the ice was assumed to be linear from top to bottom. Sea ice thickness was measured daily, and the interval of the sea ice growth rate was set to 1 day. To reduce the errors in the thickness observations, a 4-day running average was used to calculate oceanic heat flux. Figure 9 shows the calculated oceanic

heat fluxes for the three time periods (12 January–19 January; 23 January–15 February in the winter of 2009–2010; and 4 January–4 February in the winter of 2020–2021).

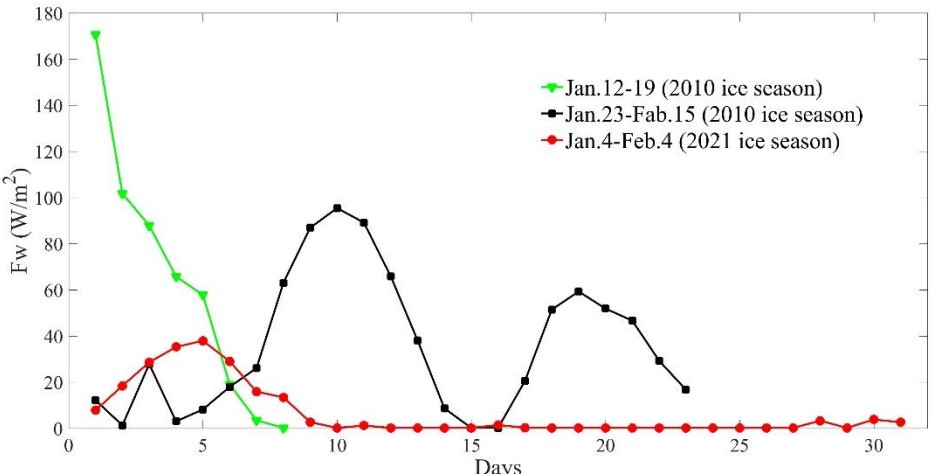

**Figure 9.** Time series of oceanic heat flux ($F_w$).

From 12 January to 17 January 2010, the oceanic heat flux decreased rapidly from a peak value of 170.6 W/m$^2$ to almost 0 W/m$^2$. From 23 January to 15 February 2010, the oceanic heat flux showed a fluctuating trend with two peaks on the 10th and 19th days, and the maximum value gradually decreased: the first maximum was 95.4 W/m$^2$, the second maximum was 59.3 W/m$^2$, and the minimum value of the oceanic heat flux was almost 0 W/m$^2$. From 4 January to 4 February 2021, the oceanic heat flux was relatively large during the first 10 days of observation and remained low thereafter, with a peak of 37.9 W/m$^2$ and a minimum value close to 0 W/m$^2$. The oceanic heat flux was averaged over three time periods. The average oceanic heat flux for the two periods in the winter of 2010 was 60 W/m$^2$ and 35 W/m$^2$, respectively. The average oceanic heat flux during the rapid growth of sea ice in the winter of 2021 was 15 W/m$^2$, and the oceanic heat flux approached 0 W/m$^2$ after the sea ice thickness stabilized.

The time series of oceanic heat flux were analyzed, which revealed that it was directly affected by both ice thickness and air temperature. When the ice thickness is the same, the intensity of the ice front at the ice–water interface increases with a decreasing air temperature, leading to stronger heat exchange between the ice and water. On the other hand, when the air temperature is the same, thicker ice results in a smaller amount of cold front energy reaching the ice–water interface, leading to a weaker energy exchange between the ice and water.

### 4.2.2. High-Resolution Thermodynamic Snow-and-Ice Model

The model requires several input parameters, including air temperature ($Ta$), wind speed ($Uz$), relative humidity ($Rh$), and solar radiation (parameterized). The initial input value is the ice thickness measured on the first day, and the calculation step size is 1 h. The values for the physical properties of sea ice are: thermal conductivity of 2.03 W/(m·K), specific heat capacity of 2093 J/(kg·K), latent heat of freezing of 333.4 kJ/kg, ice density of 917 kg/m$^3$, and freezing point of −1.4 °C. The snow was not included as an input parameter in this study because the snow was very thin in the Bohai Sea, and strong winds blew the snow away from the observation area.

The growth process of sea ice thickness was simulated using the meteorological data from the winters of 2009–2010 and 2020–2021. To investigate the influence of the oceanic heat flux on the model results, two simulations were conducted: one with a fixed oceanic heat flux of 2 W/m$^2$ and the other with the measured oceanic heat flux. The accuracy of the simulation results was evaluated by comparing them with the observation data of ice thickness.

The sea ice growth process during two time periods in 2010, 11–19 January and 23 January–15 February, was simulated, and the results are presented in Figure 10. Table 2 shows the error between the calculation results and the measured maximum ice thickness. When the oceanic heat flux was assumed to be 2 W/m², the ice thickness was significantly overestimated, and the overestimation increased over time. On 19 January, the maximum error of ice thickness was 26.1%. Similarly, when the oceanic heat flux was 2 W/m² during the second period, the overestimation of the ice thickness calculation was more severe, with an error of 60% on 15 February 2010. However, by using the oceanic heat flux calculated in Section 4.2.1 in the model calculation, the final errors of the calculation were significantly reduced to 4.2% and 0.8%, as listed in Table 2. This finding indicates that an accurate assessment of the oceanic heat flux is crucial for the accuracy of prediction during the period of rapid sea ice growth.

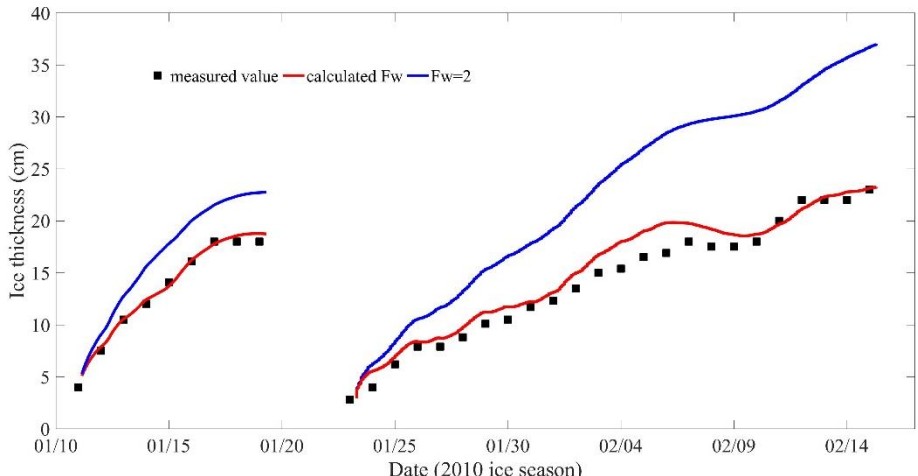

**Figure 10.** Simulation results under different oceanic heat fluxes (12 January–19 January in winter 2009–2010; 23 January–15 February in winter 2009–2010).

**Table 2.** Simulation errors at different oceanic heat fluxes.

|  | 19 January (2010 Ice Season) | | 15 February (2010 Ice Season) | | 4 February (2021 Ice Season) | |
|---|---|---|---|---|---|---|
|  | **Difference** | **Error** | **Difference** | **Error** | **Difference** | **Error** |
| $F_w = 2$ | 4.7 cm | 26.1% | 13.8 cm | 60% | 1.9 cm | 3.7% |
| Section 3.2 $F_w$ | 0.76 cm | 4.2% | 0.2 cm | 0.8% | 0.8 cm | 1.5% |

In the winter of 2010, the initial ice thickness was 0 cm, and the maximum thickness was about 20 cm. However, the 2021 ice season was different from 2010, with an initial ice thickness of 27 cm and a maximum thickness of 51 cm, as observed from 31 December 2020. The model calculations were initiated from 31 December 2020. The sea ice thickness in 2021 was much higher than that in 2010. Figure 11a shows the calculation results of ice thickness from 4 January to 4 February 2021, and the measured maximum ice thickness, along with the calculation error, is presented in Table 2. The difference between the maximum calculated value and the measured value was insignificant when the oceanic heat flux was assumed to be 2 W/m². This is because, as the ice thickness increased, the impact of the cold front at the ice–water interface was reduced, leading to a decrease in the oceanic heat flux. However, when the ice thickness was small and growing rapidly, the actual value of oceanic heat flux differed significantly from the assumed value of 2 W/m². This indicates that the oceanic heat flux has a significant impact on the accuracy of the model.

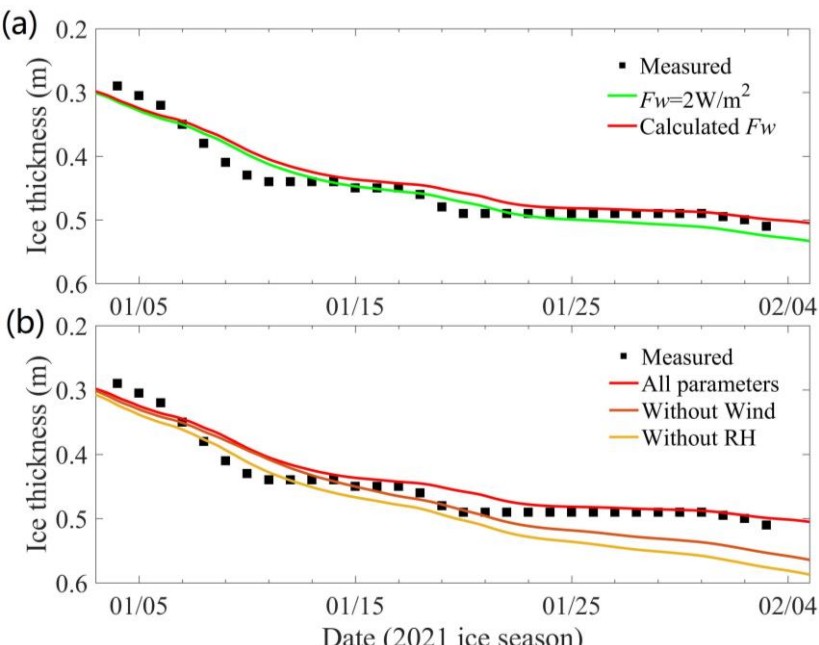

**Figure 11.** (**a**) Simulation results under different oceanic heat fluxes (4 January–4 February in 2021 ice season); (**b**) sensitivity analysis of wind and relative humidity.

### 4.2.3. Sensitivity Analysis

Sea ice growth and melting processes are primarily driven by air and hydrological factors, with air temperature being considered the most important driver of sea ice thermodynamic growth. While Stefan's law uses cumulative temperatures for sea ice thickness calculations, the sea ice numerical model adds the heat balance of the sea ice surface, heat conduction within the ice, and heat balance at the bottom of the ice, providing a more detailed understanding of the underlying physical processes. In addition to air temperature, the model also takes into account wind speed and humidity elements at the ice surface, as well as the oceanic heat flux at the bottom of the ice.

The sensitivity analysis of the oceanic heat flux to the sea ice thickness was completed in Section 4.2.2, while this section analyzes the sensitivity of wind speed and humidity. Since wind and humidity affect processes such as turbulent fluxes of sensible and latent heat, the lack of wind and humidity can result in the overestimation of the simulated ice thickness. For instance, on February 4, the ice thickness was calculated as 50.1 cm for all parameters, as 55.7 cm without wind, and as 57.9 cm without humidity, respectively. The corresponding errors compared with the measured data were 1.5%, 9.1%, and 13.6%.

Comparing Figure 11a,b, we observed that the simulated thickness error without wind and relative humidity is greater than the thickness error observed when the oceanic heat flux is set to 2 W/m$^2$. These results highlight the importance of including wind speed and relative humidity in the sea ice numerical model, especially in cases where their influence is significant.

## 5. Discussion

The state of sea ice in Liaodong Bay is influenced by both thermodynamic and dynamic factors. Due to the influence of dynamic factors, the thermodynamic observations of sea ice had not been effectively carried out in Liaodong Bay, which has hindered the development of appropriate thermodynamic models for Liaodong Bay. To address this, field observations were conducted on sea ice thickness, temperature, and wind speed in Jiangjunshi Port during the ice seasons of 2009/2010 and 2020/2021. The observed data were used to evaluate the relationship between air temperature and sea ice thickness, and to calculate the oceanic heat flux at Jiangjunshi Port. Additionally, Stefan's law and the HIGHTSI

method were applied for evaluating the sea ice growth and melting process based on the observed data.

The average oceanic heat flux during the two ice seasons of 2010 was 60 W/m² and 35 W/m², while the average oceanic heat flux during the 2021 ice season was 6.1 W/m². This trend in oceanic heat flux is consistent with the observations made by Ji [27] during the 1997/1998 ice season in Liaodong Bay on the JZ-20 platform. Ji concluded that the oceanic heat flux gradually decreased throughout the ice season and remained at 0 W/m² during the melting period. However, since the observations made in this study do not cover the melting period, calculation of the melting period cannot be provided. It is known that during the melting period, seawater temperature and sea ice temperature gradually increase, and the brine volume fraction (BVF) will increase, resulting in a decrease in latent heat [50–54]. As a result, the heat will gradually decompose the sea ice, and during the melting period, the oceanic heat flux should be greater than 0. Therefore, further studies on oceanic heat flux need to be conducted in combination with the evolution of sea ice temperature and salinity.

To investigate how temperature and ice thickness affect the oceanic heat flux in Liaodong Bay, the analysis of the relationship between ice thickness and oceanic heat flux was carried out, as shown in Figure 12a. Under the same air conditions, the difference in ice temperature distribution between thin and thick ice can affect the oceanic heat flux. It was revealed that the oceanic heat flux was higher during the rapid growth of thin sea ice, but as the sea ice thickness stabilized, the influence of cold fronts on the ice–water interface weakened, resulting in a low oceanic heat flux. Moreover, lower temperatures led to a larger temperature gradient in the ice, causing the sea ice to grow rapidly due to the cold front. It increased the discharge of salt, which in turn, increased the salinity of the seawater and continuously reduced the sea temperature. As a result, the oceanic heat flux became significantly large. Based on these findings, the relationship between air temperature and oceanic heat flux was determined (as shown in Figure 12b), and the heat flux during the rapid growth process of thin ice was calculated. Table 3 shows the statistical results.

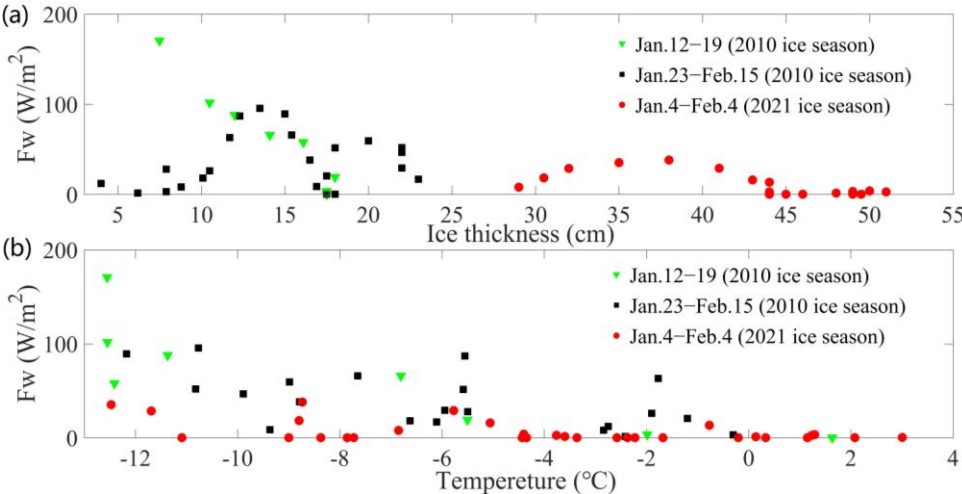

**Figure 12.** (**a**) The relationship between bottom heat flux and ice thickness; (**b**) the relationship between bottom heat flux and temperature.

**Table 3.** The oceanic heat flux during the rapid growth process of ice thickness.

| Interval Number | 1 | 2 | 3 | 4 | 5 |
|---|---|---|---|---|---|
| Temperature (°C) | −6~−4 | −8~−6 | −10~−8 | −12~−10 | −14~−12 |
| Average $F_w$ (W·m⁻²) | 35.8 | 41.6 | 38.2 | 78.4 | 104.8 |

Based on the measured air conditions and ice thickness, the growth of sea ice thickness in Jiangjunshi Port was evaluated using Stefan's law and the HIGHTSI model. The maximum error of Stefan's law was found to be over 50% using the growth coefficients from other sea areas, and thus, a recommended growth coefficient of 2.3 cm/(°C·d)$^{1/2}$ was proposed. The HIGHTSI model was used to calculate ice thickness with an oceanic heat flux of 2 W/m$^2$, and the maximum errors were 26.1%, 60%, and 5.4% in three different periods. The largest errors occurred with a small initial ice thickness. Using the oceanic heat flux evaluated in Section 4.2.1, the ice thickness errors of the HIGHTSI model were 4.2%, 0.8%, and 0.2% in the same three periods. In 1998, Cheng used sea ice observation data from Liaodong Bay to evaluate the accuracy of the HIGHTSI simulation results and found that the model accurately simulated the growth of sea ice. This paper presents the oceanic heat flux on the east coast of Liaodong Bay and verifies the results of the numerical and analytical models, providing important support for the calculation of ice thickness in the region.

## 6. Conclusions

Due to the movement of sea ice, there is a serious lack of thermodynamic observations of landfast ice in Liaodong Bay. The insufficient observation data hinder the development of sea ice thermodynamic models in this area. To address this issue, this study presented the observation works carried out in Jiangjunshi Port, with discussions on the oceanic heat flux and the models of the sea ice growth process. The following conclusions can be drawn:

(1) The daily growth of ice thickness from 0 cm decreases with the temperature, and the decrease rate is 0.26 cm/°C. The daily increase in cumulative ice thickness is influenced by air temperature and sea ice thickness. When the sea ice thickness reaches 20 cm, the growth rate decreases to around 1 cm/d.

(2) The error of Stefan's law and the HIGHTSI model was evaluated. It was found that a growth coefficient of 2.3 cm/(°C·d)$^{1/2}$ is more consistent with the measured value. Meanwhile, the HIGHTSI model is strongly dependent on the oceanic heat flux value when the ice is thin.

(3) The residual method was used to calculate the oceanic heat flux of Jiangjunshi Port. The average oceanic heat flux in the first period of the 2010 ice season was 60 W/m$^2$, and in the second period, it was 35 W/m$^2$. In the ice season of 2021, the average oceanic heat flux was 6.1 W/m$^2$.

Nowadays, global climate change has resulted in unpredictable changes in ice conditions in Liaodong Bay, which has a significant impact on human economic activities. Due to the limitations in observation conditions, numerical simulations of sea ice have become an important approach for the continuous evaluation of the ice conditions in Liaodong Bay. Therefore, the work carried out in this study can provide valuable data for assessing the ice thickness here. However, this study also has some limitations, including a lack of observations of sea ice temperature and salinity. Considering more physical and mechanical properties of sea ice in field experiments would provide a more optimized and accurate thermodynamic model for Liaodong Bay.

**Author Contributions:** N.X. and D.D. initiated this work; Y.M., S.Y. and W.S. conducted field observations; Y.M. and W.S. carried out modeling experiments, analyzed the results, and wrote the original draft; N.X. and S.Y. collected the data. All the authors contributed to the writing and editing of the manuscript. All authors have read and agreed to the published version of the manuscript.

**Funding:** This research was funded by the National Natural Science Foundation of China, grant number 42206221; Laoshan Laboratory, grant number LSKJ 202203900; National Natural Science Foundation of China, grant number U1806214; State Key Laboratory of Coastal and Offshore Engineering of Dalian University of Technology the Open Fund; and State Environmental Protection Key Laboratory of Coastal Ecosystem.

**Data Availability Statement:** The data presented in this study are available on request from the corresponding author. The data are not publicly available due to company data secrecy provision.

**Conflicts of Interest:** The authors declare no conflict of interest.

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
