# Peer review of "Ice Mass Balance in Liaodong Bay: Modeling and Observations"

_water, doi:10.3390/w15050943_

Round 1
Reviewer 1 Report
1. Typos:
- Line 12: “A In the winters of”
- Line 14: “researh” – “research”
- Line 24 and Keywords: “analytic mode” – “analytic model”
- Line 89: “blockedthe” – “blocked the”
- Line 95: “Atmospheric Condition” – “atmospheric condition”
- Line 126: “datawere” – “data were”
- Line 234: “is setting to” – “is set to”
- Line 259: “Stefan’s Law of Ice Growth” – “Stefan’s law of ice growth”
- Line 403: “werelarge” – “were large”
- “the increasing of ” – “the increase of”
2. The full name of some abbreviation must be provided at the beginning.:
- Line 15: The definition of DIT must be provided, although it seems to be related with “decreasing”.
- Line 53: the detailed name of HIGHTSO model should be added at the beginning, not line 303.
3. the abstract should be examined again for typos, clarity, and conciseness
- Line 82: Before the air-ice-water observation in the Liaodong Bay was carried out.
- Line 78: “2.1 Study Area” – better expression by “Observation area” or “Scope of observation”?
- Line 152: “In this paper, the HIGHTSI model was used to simulate …”
- The “First Person” is suggested to be replaced by the “Third person”.
4. Comments on figures.
- Fig. 1 is suggested to be updated with a map of larger scope, so that the readers from other regions can better understand the location.
- Figs. 2, 3, 5: the figures of higher resolution are recommended.
- Fig. 4, the definition of “dh” and “dt” should be right near this figure.
- Fig. 6: The definition of “Fw” should be provided.
5. For better clarity and conciseness of the “conclusion part”, it is suggested to split the “conclusion part” into several paragraphs or several bullet items.
6. There is no snowy data in the input value of Table 1. Did the influence of snowy ignore? When these are disregarded, what is the reason?
7. What are the initial parameters of the calculation?
8. The background and motivation of the work need to be clarified.

Author Response
- Typos:
- Line 12: “A In the winters of”
A: Done, thank you.
- Line 14: “researh” – “research”
A: Done, thank you.
- Line 24 and Keywords: “analytic mode” – “analytic model”
A: Done, thank you.
- Line 89: “blockedthe” – “blocked the”
A: Done, thank you.
- Line 95: “Atmospheric Condition” – “atmospheric condition”
A: Done, thank you.
- Line 126: “datawere” – “data were”
A: Done, thank you.
- Line 234: “is setting to” – “is set to”
A: Done, thank you.
- Line 259: “Stefan’s Law of Ice Growth” – “Stefan’s law of ice growth”
A: Done, thank you.
- Line 403: “werelarge” – “were large”
A: Done, thank you.
- “the increasing of ” – “the increase of”
A: Done, thank you.
- The full name of some abbreviation must be provided at the beginning.:
- Line 15: The definition of DIT must be provided, although it seems to be related with “decreasing”.
A: The daily ice thickness is DIT, thank you.
- Line 53: the detailed name of HIGHTSO model should be added at the beginning, not line 303.
A: Done, thank you.
- the abstract should be examined again for typos, clarity, and conciseness
- Line 82: Before the air-ice-water observation in the Liaodong Bay was carried out.
A: Done, thank you.
- Line 78: “2.1 Study Area” – better expression by “Observation area” or “Scope of observation”?
A: Done, thank you.
- Line 152: “In this paper, the HIGHTSI model was used to simulate …”
A: Done, thank you.
- The “First Person” is suggested to be replaced by the “Third person”.
A: Done, thank you.
- Comments on figures.
- 1 is suggested to be updated with a map of larger scope, so that the readers from other regions can better understand the location.
A: Done, thank you.
- 2, 3, 5: the figures of higher resolution are recommended.
A: Done, thank you.
- 4, the definition of “dh” and “dt” should be right near this figure.
A: Done, thank you.
- 6: The definition of “Fw” should be provided.
A: Done, is oceanic heat flux. Thank you.
- For better clarity and conciseness of the “conclusion part”, it is suggested to split the “conclusion part” into several paragraphs or several bullet items.
A: Done, We have divided the results into three parts. Thank you.
- There is no snowy data in the input value of Table 1. Did the influence of snowy ignore? When these are disregarded, what is the reason?
A: The HIGHTSI model is entitled as “High-resolution Thermodynamic Snow and Ice model”, so snow thermodynamics is an important component of HIGHTSI (e.g., Launiainen and Cheng, 1998, Cheng et al., 2003 and 2008). However, for this case study, snow was not considered. Because snow was very thin on top of the Bohai Sea ice, and strong wind make the snow away from the port, therefore, we didn’t add snow physics in paper. Thank you.
- What are the initial parameters of the calculation?
A: The main initial parameter of the mode is ice thickness, and we use the measured ice thickness of the first day as the initial ice thickness. Thank you.
- The background and motivation of the work need to be clarified.
A: We have updated the introduction. Thank you.

Reviewer 2 Report
The authors have made efforts to understand ice mass balance in liaodong Bay through filed observations. However, the paper needs major modifications in terms of statistical analysis and effective parameters:
1-Statistical analysis is poorly written for modeling ice balance mass analysis
2-Literature review requires research works between 2000-2023
3-What are effective parameters on the modeling ice balance mass. A new section should be considered with a lot of justifications. Furthermore, a sensitivity analysis can be done for this purpose.
4-Which software was used to model this phenomenon. A new section should be considered with a lot of justifications.
5-Literature review can improve by using: Receiving More Accurate Predictions for Longitudinal Dispersion Coefficients in Water Pipelines: Training Group Method of Data Handling Using Extreme Learning Machine Conceptions
6-Introduction section needs improvement of motivation of this work, innovation, and Research organization.
Author Response
1-Statistical analysis is poorly written for modeling ice balance mass analysis
A: We have added section 3.2 to describe statistical methods, and supplementing the value of R2. Based on the measured data, the sea ice freezing rate under Stefan's law was statistically analyzed. We have added Section 4.2.3 for sensitivity analysis. Thank you.
2-Literature review requires research works between 2000-2023.
A: We have updated the introduction with recent research. Thank you.
3-What are effective parameters on the modeling ice balance mass. A new section should be considered with a lot of justifications. Furthermore, a sensitivity analysis can be done for this purpose.
A: Temperature is the most important factor, and the simulated thickness error without wind and relative humidity is greater than thickness error which the oceanic heat flux is 2 W/m2. We have added Section 4.2.3 for sensitivity analysis. Thank you.
4-Which software was used to model this phenomenon. A new section should be considered with a lot of justifications.
A: The paper use the HIGHTSI model, we add a description of HIGHTSI in Section 3.1.2. Thank you.
5-Literature review can improve by using: Receiving More Accurate Predictions for Longitudinal Dispersion Coefficients in Water: Training Group Method of Data Handling Using Extreme Learning Machine Conceptions
A: We have updated the introduction, and add studies that used data learning to predict sea ice extent. And “the water in ice” was parameterized as the oceanic heat flux (Fw). Thank you.
6-Introduction section needs improvement of motivation of this work, innovation, and Research organization.
A: We have updated the introduction, and improved the motivation and innovation of this article. We have reorganized the chapters of the manuscript. Thank you.

Reviewer 3 Report
Hereby, thank you very much for the opportunity to review an article that describes a very important global problem - so far in the local space.
Hereby, thank you very much for the opportunity to review an article that describes a very important global problem - so far in the local space.
I have included the comments in the attached manuscript.
However, I would like to add that the article is written very clearly.
I consider the topic discussed to be scientifically significant - and I hope that in the future the authors will develop the researched area.
As for comments - I am posting a list below.
In general:
1. in equations change big bold multiply sine.
2. At the begining define whats HIGHTSI model
3. (°C•d) -1/2 – please explain it to the readers. Specialist know.
4. I like Figure 1. But i am affraid that not everyone would know where is it on world map.
5. I am asking for information about the values ​​of coefficients in linear equations
appearing in the manuscript
6. Table 3 – Heading would be nice
I recommend the article for publication - after minor changes in the Water magazine.

Author Response
1.in equations change big bold multiply sine.
A: We have unified the format of equations, thank you.
2.At the begining define whats HIGHTSI model
A: Done, we add a description of HIGHTSI in Section 3.1.2. Thank you.
3.(°C•d) -1/2 – please explain it to the readers. Specialist know.
A: Done, the unit of a is calculated from the units of , , and . Thank you.
4.I like Figure 1. But i am affraid that not everyone would know where is it on world map.
A: We update the map of larger scope. Thank you.
5.I am asking for information about the values of coefficients in linear equations appearing in the manuscript?
A: We have added section 3.2 to describe statistical methods, and supplementing the value of R2. The information of the coefficient is the rate of sea ice growth, and we added analysis of freezing rate. Thank you.
6.Table 3 – Heading would be nice
Done, Thank you.

Round 2
Reviewer 2 Report
Accept as is
Author Response
Thank you for your suggestion.
Best regards,
Yuxian Ma